# Nicotine Administration Augments Abdominal Aortic Aneurysm Progression in Rats

**DOI:** 10.3390/biomedicines11051417

**Published:** 2023-05-10

**Authors:** Hana Hadzikadunic, Tea Bøvling Sjælland, Jes S. Lindholt, Lasse Bach Steffensen, Hans Christian Beck, Egle Kavaliunaite, Lars Melholt Rasmussen, Jane Stubbe

**Affiliations:** 1Elitary Research Centre of Individualized Treatment for Arterial Disease (CIMA), Odense University Hospital, University of Southern Denmark, 5000 Odense, Denmark; 2Cardiovascular and Renal Research Unit, Institute for Molecular Medicine, University of Southern Denmark, 5000 Odense, Denmark; 3Department of Cardiothoracic and Vascular Surgery, Odense University Hospital, 5000 Odense, Denmark; 4Department of Clinical Biochemistry, Odense University Hospital, 5000 Odense, Denmark

**Keywords:** aortic aneurysm, alpha7 nicotinic acetylcholine receptor, inflammation, vascular remodeling, therapeutic strategy, animal model, nicotine

## Abstract

Inflammation and elastin degradation are key hallmarks in the pathogenesis of abdominal aortic aneurysms (AAAs). It has been acknowledged that activation of alpha7 nicotinic acetylcholine receptors (α7nAChRs) attenuates inflammation, termed the cholinergic anti-inflammatory pathway (CAP). Thus, we hypothesize that low-dose nicotine impairs the progression of elastase-induced AAAs in rats by exerting anti-inflammatory and anti-oxidative stress properties. Male Sprague–Dawley rats underwent surgical AAA induction with intraluminal elastase infusion. We compared vehicle rats with rats treated with nicotine (1.25 mg/kg/day), and aneurysm progression was monitored by weekly ultrasound images for 28 days. Nicotine treatment significantly promoted AAA progression (*p* = 0.031). Additionally, gelatin zymography demonstrated that nicotine significantly reduced pro-matrix metalloproteinase (pro-MMP) 2 (*p* = 0.029) and MMP9 (*p* = 0.030) activity in aneurysmal tissue. No significant difference was found in the elastin content or the score of elastin degradation between the groups. Neither infiltrating neutrophils nor macrophages, nor aneurysmal messenger RNA (mRNA) levels of pro- or anti-inflammatory cytokines, differed between the vehicle and nicotine groups. Finally, no difference in mRNA levels of markers for anti-oxidative stress or the vascular smooth muscle cells’ contractile phenotype was observed. However, proteomics analyses of non-aneurysmal abdominal aortas revealed that nicotine decreased myristoylated alanine-rich C-kinase substrate and proteins, in ontology terms, inflammatory response and reactive oxygen species, and in contradiction to augmented AAAs. In conclusion, nicotine at a dose of 1.25 mg/kg/day augments AAA expansion in this elastase AAA model. These results do not support the use of low-dose nicotine administration for the prevention of AAA progression.

## 1. Introduction

Abdominal aortic aneurysms (AAAs) refer to a permanent, localized dilatation of the abdominal aorta in which the aortic diameter is >3.0 cm or 1.5 times the expected normal diameter [1]. AAAs are more common in men with an estimated prevalence between 1% and 5% in men aged ≥65 years [2]. Rupture of AAAs is a life-threatening event and is globally estimated to lead to 200,000 deaths annually, with higher rupture risk as the size of the AAA increases [3,4]. The mortality rate associated with rupture is 85–90% [4]. Currently, the only established treatment to prevent rupture is surgical repair [5]. However, randomized clinical trials have shown that surgical AAA repair is not beneficial in smaller AAAs [6,7]. Therefore, clinical guidelines recommend that AAAs < 5.5 cm should be managed by interval surveillance imaging until the diameter reaches the surgical threshold of 5.5 cm [5,6]. These surgical procedures are both costly and not risk free [6]; thus, an unmet clinical need exists to develop novel medical therapies for AAAs < 5.5 cm to prevent aneurysm progression, the need for surgical repair and rupture [7].

Pathologically, AAAs are considered a multifactorial process with inflammation as a central hallmark [8]. Aortic inflammation is characterized by infiltration of pro-inflammatory immune cells, including macrophages and neutrophils [9]. Upon infiltration, these cells secrete pro-inflammatory cytokines and matrix metalloproteinases (MMPs) which, upon activation, continue degradation of elastin in the medial layer of the aortic wall, leading to a reorganization of the aortic wall extracellular matrix (ECM) [10]. The continuous degradation and remodeling ultimately weaken the aortic wall, with subsequent expansion of the aorta and eventually rupture of the aneurysm [9,10]. Furthermore, aortic inflammation is associated with local production of reactive oxygen species (ROS) that further augments degradation of the ECM by increasing MMP production, resulting in aneurysm progression [11,12]. Therefore, anti-inflammatory therapies could have the potential of addressing the unmet clinical need to treat unruptured AAAs [13].

Although smoking is one of the well-recognized risk factors for the development of AAAs [14], recent studies have reported that nicotine, the major component of cigarettes, can attenuate inflammation by selectively inhibiting the synthesis of pro-inflammatory cytokines [15,16,17]. This anti-inflammatory mechanism is partly mediated through the vagus nerve by activating alpha7 nicotinic acetylcholine receptors (α7nAChRs) and directly via α7nAChRs on immune cells and cells in the aortic wall. This receptor is a ligand-gated ion channel and is composed of five α7 subunits that, upon activation, initiate the cholinergic anti-inflammatory pathway (CAP) [18]. Several preclinical studies have shown that activation of CAP attenuates inflammatory conditions, including arthritis, ulcerative colitis, and sepsis [19,20,21]. Furthermore, in a murine model of AAA, the activation of α7nAChRs by a cholinergic agonist prevented AAA progression by a mechanism associated with the suppression of pro-inflammatory cytokines and the inhibition of vascular smooth muscle cell pyroptosis, mediated by inflammasome 3 and thereby a reduction in MMP activity [22]. Thus, accumulating evidence supports the notion that the stimulation of the CAP by activating α7nAChRs can exert protective, anti-inflammatory properties. This makes the activation of α7nAChRs a potential therapeutic target for future medical treatment of AAAs [23]. However, the effects of nicotine as a cholinergic agonist remain contradictory in cardiovascular diseases and appear to work in a dose-dependent manner [24].

In this study, we hypothesize that low-dose nicotine impairs the progression of elastase-induced AAAs in male rats by exerting anti-inflammatory and anti-oxidative stress properties.

## 2. Materials and Methods

### 2.1. Ethical Statements 

All procedures in this study were performed in accordance with the experimental protocol approved by the Danish Animal Experiments Inspectorate (permission number: 2020-15-0201-00499). Animal care conformed to ARRIVE guidelines [25]. As females are generally protected against AAA formation [26,27,28,29], we only used male rats in this study.

The sample size calculation was based on our previous studies with the elastase AAA model described by Melin et al. [30] and was set to 12 rats per group. Due to the conservative sample size estimation, repeated-measures analysis of variance (ANOVA) was used. 

### 2.2. Experimental Animals, Housing, and Husbandry 

Male Sprague–Dawley rats (300 g) were purchased from Janvier Laboratories (Le Genest-Saint-Isle, France). Upon arrival, rats were allowed 5 days to acclimatize prior to inclusion in the experiments. All animals were housed with up to 4 rats per cage at the Biomedical Laboratory, University of Southern Denmark, in a 12 h light/dark cycle, room temperature of 20 °C and air humidity of 55%, with standard chow and water available ad libitum throughout the experiment. 

### 2.3. Study Design 

In this study, a total of 32 male rats entered the surgical protocol with 4 investigators conducting all surgical procedures blinded to treatment. Rats were only included in the study if the surgical procedure was successfully performed with the continuous absence of humane endpoints following surgery. This led to the exclusion of 5 rats before initiation of treatment on day 1 after surgery. An extern investigator blinded to the surgical procedures randomly allocated rats into the vehicle group (*n* = 14) or the nicotine group (*n* = 13). 

The day after surgical AAA induction, the nicotine group received a dose of 1.25 mg/kg/day, administered by subcutaneous injection, while the vehicle group received an equivalent volume of sterile physiological saline (1.1–1.7 mL, Fresenius Kabi, Copenhagen, Denmark). Body weights were monitored daily throughout the experimental period. 

Investigators were blinded to treatment allocation during the conduct of all experiments, outcome assessment and data analyses. 

### 2.4. Porcine Pancreatic- Elastase-Induced AAA Model 

During the acclimatization period, the rats were habituated to ingesting nut paste in preparation for sufficient pain management during the following operative course. One hour prior to anesthesia, each rat was administered temgesic (buprenorphine) 0.2 mg (#563789; Orifarm, Odense, Denmark) in 1 g nut paste [31]. Then, all rats were anesthetized, and the surgical procedure for infrarenal perfusion of porcine pancreatic elastase (Sigma-Aldrich, Søborg, Denmark) for 30 min was initiated as described by Melin et al. [30] (Figure 1); the only exception was that rats were infused with 10 unit/mL for 30 min. 

Post-operatively, each rat was once again administered temgesic 0.2 mg in 1 g nut paste upon awakening for pain relief [31] and placed in a heating cabinet overnight for recovery. The pain relief protocol was repeated the following day [31]. 

### 2.5. Nicotine Preparation and Dosage 

Nicotine hydrogen tartrate salt (Sigma-Aldrich, Søborg, Denmark) was dissolved in 0.9% physiological saline (Fresenius Kabi, Copenhagen, Denmark) to obtain a concentration of 0.42 μg/mL and adjusted to pH 7.2 with NaOH and then sterile-filtrated. The stock solution was stored at −20 °C and thawed just prior to administration. The dose of nicotine chosen for this study was 1.25 mg/kg/day and reported as nicotine hydrogen tartrate salt (corresponding to 0.406 mg/kg/day nicotine-free base) [32]. 

### 2.6. Measuring AAA Progression with Ultrasound Imaging 

All ultrasound recordings (LogiQe ultrasound machine and a L10-22-RS transducer, GE Healthcare, Chalfont Saint Giles, UK) were performed in the axial plane from the renal arteries to bifurcation at day 0 prior to surgery and at post-surgery at day 7, 14, 21 and 28 as described by Melin et al. [30]. The maximal anterior–posterior diameter and circumference of the aneurysm during systole were measured in all rats. The circumference of the aneurysm was used to assess the relative AAA increase in percentage. This value was obtained by adjusting to the baseline aortic size recorded on day 0. 

### 2.7. Euthanization and Tissue Preparation

All rats were euthanized post-surgery on day 28. Organs including the heart, spleen, lungs, and kidneys were harvested and weighted for organomegaly. The upper part of aneurysms and unaffected abdominal aorta proximal of the renal arteries were harvested from all rats, fixed in 10% normal formalin-buffered saline (Hounisen Laboratorieudstyr A/S, Skanderborg, Denmark) for 24 h and afterward washed in 0.05% sodium azide in phosphate-buffered saline (PBS, Thermo Fisher, Waltham, MA, USA) at 4ºC until embedding in paraffin for later histological analyses. The lower part of aneurysms and unaffected abdominal aortas were snap-frozen in liquid nitrogen and stored at −80 °C for later RNA isolation, proteomics and zymography.

### 2.8. Histological and Immunohistochemical Analyses 

Five µm cross-sections of the abdominal aorta and the aneurysm were used for detection of elastin by Miller’s elastin staining (Atom Scientific, Hyde, UK) according to the manufacturer’s instructions. Other aneurysm cross-sections were used for detection of neutrophils (myeloperoxidase, MPO, 1:8000, ab188211, Abcam, Cambridge, UK) and macrophage marker Cluster of Differentiation 68 (CD68, 1:500, ab125212, Abcam, Cambridge, UK) using the protocol previously described by Melin et al. [30]. For both labeling protocols, rabbit immunoglobulin (X0903, DAKO, Agilent, Glostrup, Denmark) in the corresponding concentration was used as a negative control.

Micrographs were captured using an Olympus Bx51 microscope with an attached Olympus DP26 camera. ImageJ software (ImageJ 1.53a Wayne Rasband, National Institutes of Health, Bethesda, MD, USA) was used to quantify histological stainings and identify tunica media. The percentage of elastin remaining in tunica media was quantified with a color threshold tool. In addition, the degree of elastin degradation in tunica media in Miller’s stained sections were divided into 8 areas and scored from 1–4, 1 being organized elastin with no ruptures and 4 being the total disruption of complete concentric elastin lamellae by 2 investigators. The individual variation was 5.67%; thus, the average score was used for analysis. 

### 2.9. Quantitative Reverse Transcriptase-Polymerase Chain Reaction (qRT-PCR)

Aneurysmal total RNA was isolated using the trizol method (Ambion), and cDNA synthesis was performed using iScript (Biorad, Copenhagen, Denmark). Messenger RNA (mRNA) levels were determined by qPCR using SYBR green for detection and primers as described by Melin et al. [30]. Additional primer sequences for α-actin included 5′-GTC ATG TCA GGG GGC ACT AC-3′, 5′-ACA TCT GCT GGA AGG TGG AC-3′, myocardin 5′-ACT GAGTTC CAT GAC CCG AG-3′; 5′-ATG GAT CTT TCT GCC GTG GA-3′and osteopontin 5′-TGA TGA ACA GTA TCC CAT G-3′, 5′-AAC TGG GAT GAC CTT GAT AG-3′. All samples were run in duplicate with water and total RNA used as negative controls. All mRNA levels were normalized to housekeeping gene Ribosomal Protein L41 (RPL41) and displayed as times of the vehicle-treated group.

### 2.10. Zymography

Proteins were extracted by homogenization in an extraction buffer. The bicinchoninic acid protein determination kit (Sigma-Aldrich, Søborg, Denmark) was used to determine protein concentrations using serum albumin as standard. AAA protein samples (12 µg) and 1.25 µL recombinant MMP2 (Sigma Aldrich, Søborg, Denmark) were loaded onto Novex zymogram gel, separated, renatured, and incubated for 24 h at 37 °C in a developing buffer, before undigested proteins were stained by a simple blue stain (Thermo Fischer, Waltham, MA, USA). White bands representing digested proteins were inverted and quantified in Molecular Imager Image Lab (ChemiDoc WRS+, Biorad, Copenhagen, Denmark) as described by Melin et al. [30].

### 2.11. Proteomic Analysis 

Preparation of non-aneurysmal abdominal aortic tissue for mass spectrometry was performed virtually as previously described [33]. Briefly, formalin-fixed tissue was homogenized in a lysis buffer, denatured, alkylated, and subjected to overnight digestion with trypsin. Tryptic peptide samples were purified and tagged with 16-plex tandem mass tags (TMT, Thermo Scientific, Waltham, MA, USA) using mass tag 126 as the internal control (pool of all samples). Proteome data were protein abundances relative to the internal control. Mixed peptide samples were fractionated by high-pH chromatography followed by nano-LC–MS/MS analysis, virtually, as previously described [34] with the following modifications: Samples were analyzed using FAIMS Pro interface (Thermo Fischer Scientific, Slangerup, Denmark) and mass spectra were acquired by switching between CVs of −50 V and −70 V with a 2 s cycle time. All Eclipse raw data files were processed and quantified using Proteome Discoverer version 2.4 (Thermo Scientific, Waltham, MA, USA), also as previously described [34]. 

### 2.12. Statistical Analyses 

The Shapiro–Wilk normality test was applied to examine if data were normally distributed, and appropriate statistical methods were chosen according to the outcome. A parametric test was used when data were normally distributed and presented as mean ± SD. When data failed normality testing, a non-parametric test was used, thus presenting the data as median ± interquartile range (IQR). *p*-values < 0.05 were considered significant. 

The difference in relative AAA increase between the vehicle group and the nicotine group was analyzed using mixed-repeated-measures ANOVA. When data failed to meet the assumption of sphericity performed with Mauchly’s test, either a Greenhouse–Geisser or Huynh–Feldt correction was used depending on the values of epsilon. When significance was determined following mixed-repeated-measures ANOVA, Sidak’s multiple-comparison test was used in the post-hoc analysis to determine the difference between groups. 

A parametric independent two-way Student’s *t*-test was used to compare the means between the groups regarding the percentage of elastin remaining in tunica media, the grade of elastin degradation evaluated in 8 areas of the aneurysm and body weight at entry and on day 28, mRNA levels and zymography. When the F-test for comparison of variation between groups was significant, Students *t*-test with Welch’s correction was used and indicated in relevant figure legends.

A non-parametric Mann–Whitney U test was used to detect a difference in all histological and immunohistochemical quantifications among the two groups.

Data from proteomic analyses were analyzed by Student’s *t*-test for each protein and subsequent false discovery rate (FDR) correction for multiple testing. Enrichment analyses were performed using the clusterProfiler 4.0 package [35] in R based on ranked log2-transformed fold changes. Default settings were used for gseGO and gseKEGG functions. No enrichment was found after correcting for multiple testing by the Benjamini–Hochberg method, so *p*-values were not adjusted in the presented data. 

Statistical analyses were performed using SPSS (IBM SPSS Statistic for MAC, version 28.0, IBM Corp., Armonk, NY, USA) or GraphPad Prism software (version 6.07, GraphPad Software Inc., Boston, MA, USA). All graphs were constructed using GraphPad Prism software. 

## 3. Results

All rats entered the experimental protocol with no difference in body weight at the day of surgery (vehicle: 376.0 ± 9.6 g, nicotine: 377.6 ± 8.9 g, *p* = 0.650, *n* = 14/13). Rats in both groups gained appropriate weight during the study period with no difference in body weight at the end of the experiment between the groups (vehicle: 492.4 ± 34.0 g, nicotine: 487.7 ± 29.2 g, *p* = 0.702, *n* = 14/13). Additionally, no changes in organ-to-body weight ratios in terms of the heart, spleen, lungs, and kidneys were found between the groups on day 28 (Table 1).

Data from the heart and spleen are presented as median ± interquartile range, while data from lungs and kidneys are presented as mean ± standard deviation. 

### 3.1. Nicotine Treatment Promoted AAA Progression

The increase in maximal AAA size was not statistically different on day 7 (Figure 2, *p* = 0.883) or on day 14 (Figure 2, *p* = 0.093) between the vehicle and nicotine-treated groups. In contrast, we found that nicotine treatment significantly increased the AAA size on day 21 (Figure 2, *p* = 0.028) and day 28 (Figure 2, *p* = 0.031) compared to the vehicle-treated group. These results indicate that nicotine treatment in the given dose promotes AAA formation in our rat AAA model.

### 3.2. Nicotine Treatment Did Not Alter the Elastin Content or Architecture in Tunica Media

To investigate the aneurysmal elastin structure, we first assessed the percentage of elastin remaining in tunica media after the experimental period of 28 days (Figure 3A,B). No difference in percentage of elastin remaining in tunica media between the groups was found (Figure 3B, *p* = 0.969). In both groups, we found irregular elastin degradation in the aneurysmal cross-section. Scoring the elastin degradation in 8 areas of the aneurysm cross-sections, we found well-preserved elastin architecture in some areas (grade 1), while other areas contained extensive defragmentation of elastic lamellae with an unidentifiable border zone between tunica media and adventitia (grade 4) (Figure 3C,D). However, there was no significant difference in the score of elastin degradation between the vehicle and nicotine-treated groups (Figure 3E, *p* = 0.255). Overall, these results suggest that nicotine treatment does not affect elastin integrity. 

### 3.3. Nicotine Treatment Suppressed Pro-MMP2 and MMP9 Activity

We examined the effect of nicotine on mRNA levels of MMPs and MMP activity by zymography to determine whether nicotine treatment has a protective effect against elastin degradation. No significant difference in mRNA levels of MMP2 (*p* = 0.479) or MMP9 (*p* = 0.872) was observed between the vehicle and nicotine-treated groups (Figure 4A,B). In contrast, we found significantly decreased pro-MMP2 (*p* = 0.029) and MMP9 (*p* = 0.030) activity measured by zymography in the nicotine group (Figure 4C–E). Meanwhile, MMP2 activity showed a trend to be decreased though non-significantly by nicotine (Figure 4C,F, *p* = 0.092). Furthermore, the number of MMP9-positive cells were determined in aneurysmal cross-sections in both the vehicle and nicotine group (Figure 4G). There was no significant difference in MMP9-positive cells per mm^2^ between the two groups (Figure 4H, *p* = 0.396). Based on these data, we demonstrated that nicotine treatment partially exerts protective effects on the degradation of elastin by reducing pro-MMP2 and MMP9 activity. 

### 3.4. Nicotine Treatment Failed to Exert Immunoregulatory Effects on Infiltration of Inflammatory Cells 

To determine the anti-inflammatory effect of nicotine, we first examined the number of MPO and CD68-positive cells in aneurysmal cross-sections in the two groups (Figure 5A,B). Visually, MPO and CD68-positive cells were primary located in tunica intima and tunica adventitia with few migrating cells in tunica media. When semi-quantified, there was no significant difference in MPO-positive cells per mm^2^ between the vehicle and nicotine-treated groups (Figure 5C, *p* = 0.928). There was also no difference found in CD68-positive cells per mm^2^ between the groups (Figure 5D, *p* = 0.975). 

Next, we determined the mRNA levels of pro-inflammatory and anti-inflammatory cytokines. Neither IL6 (*p* = 0.925), TNF (*p* = 0.791) or IL10 (*p* = 0.104) were affected by nicotine treatment (Figure 5E–G). Thus, our results did not show anti-inflammatory properties of nicotine treatment.

### 3.5. Nicotine Treatment Did Not Reduce Oxidative Stress 

We next looked at the impact of nicotine on aneurysmal oxidative stress levels by determining mRNA levels of the potent antioxidants Nrf2 and HO-1 in aneurysmal tissue. No difference in Nrf2 (*p* = 0.455) or HO-1 (*p* = 0.163) mRNA levels was shown between the vehicle and nicotine group (Figure 6). These results indicate that nicotine treatment does not attenuate oxidative stress in this elastase AAA model. 

### 3.6. Nicotine Treatment Did Not Affect Vascular Smooth Muscle Cells 

To investigate the effect of nicotine on vascular smooth muscle cells in the aneurysm wall, mRNA levels of α-actin, myocardin and osteopontin were determined. Nicotine treatment neither affected mRNA levels of α-actin (Figure 7A, *p* = 0.289), myocardin (Figure 7B, *p* = 0.744) or osteopontin (Figure 7C, *p* = 0.395). These results suggests that low-dose nicotine treatment does not protect vascular smooth muscle cells in aneurysmal tissue.

### 3.7. Proteomic Analyses of Abdominal Aorta Reveal That Nicotine Treatment Reduced Inflammatory Response and Cellular Response to Reactive Oxygen Species 

To assess the impact of nicotine on the abdominal aortic wall in the absence of aneurysms and potentially reveal mechanisms driving toward larger aneurysm development by nicotine treatment, we conducted proteomic analysis on the abdominal aorta of rats. We compared the results from nicotine-treated rats (*n* = 12) to those from untreated rats (*n* = 13). Our analysis revealed that out of the 2264 proteins identified, 50 (2.2%) were nominally upregulated and 234 (10.3%) were downregulated (unadjusted *p* < 0.05). However, only three proteins were significantly downregulated post-correction for multiple testing: the actin-binding protein Tropomodulin-2 (Tmod2), known to stabilize F-actin in neurons [36]; the cytoskeletal filamentous F-actin Myristoylated alanine-rich C-kinase substrate (MARCKS); and the plasma-membrane-anchored protein Paralemmin-1 (Palm) associated with fibroblast, neurons, and lymphatic endothelial cells (Appendix A). 

To test the hypothesis that nicotine treatment causes anti-inflammatory or anti-oxidative stress on the aortic wall, we investigated gene ontology terms “inflammatory response” (GO:0006954) and “cellular response to reactive oxygen species” (GO:0034614) (Figure 8A,B). Both terms were significantly downregulated (*p* = 0.02 and *p* < 0.00001, *χ^2^* test). To further interrogate the effect of nicotine on non-aneurysmal abdominal aorta, we performed unbiased enrichment analysis based on the GO and KEGG databases. 

When corrected for multiple testing, no pathways were significantly regulated (q < 0.05, Appendix A). However, several pathways were nominally regulated in nicotine-treated rat aortas. These include the lipoxygenase pathway and pyrimidine metabolic processes and the Ras signaling pathway, which are amongst the upregulated pathways, while fatty acid metabolism, cell adhesion and the IL17 signaling pathway are downregulated, which could potentially contribute to the understanding of the nicotine effect on abdominal aorta (Figure 8C,D). 

## 4. Discussion

This study shows that nicotine at a dose of 1.25 mg/kg/day significantly promoted the progression of AAAs after 27 days of subcutaneous treatment. This increase in AAA growth was not associated with changes in elastin integrity, changes in infiltrating neutrophils nor macrophages, nor changes in aneurysmal mRNA levels of the inflammatory cytokines TNF, IL6 and IL10. Although, nicotine treatment did result in a decrease in MMP9 activity in the aneurysm wall. In contrast, the proteomic data from non-aneurysmal abdominal aorta showed downregulation of proteins MARCKS, Tmod2 and Palm and proteins related to ontology terms: Inflammation and oxidative stress. 

Unexpectedly, the progression of AAAs was inconsistent with our hypothesis. As nicotine is a cholinergic agonist with the potential of targeting the CAP through the activation of α7nAChRs [37], we anticipated that treatment with low-dose nicotine would suppress AAA growth, as shown for other α7nAChRs agonists [22,38]. In support of our hypothesis, recent data from the DANCAVAS studies found that smoking, where nicotine inhalation is a major component, appeared to be protective against the development of ascending aortic aneurysms [39]; thus, we speculated it was also likely to be protective against AAAs. In support of this, low-dose nicotine (10^−8^ M) reduced the inflammatory response to LPS stimulation in human monocytes at transcriptional levels in vitro [40]. 

Furthermore, our proteomic data from non-aneurysmal abdominal aorta reveals that proteins involved in inflammatory response, and reactive oxygen species, in ontology terms, were significantly downregulated. In line with these pathways, the pro-inflammatory IL17 signaling pathway [41] and cell adhesion molecules, in KEGG terms, were also downregulated by nicotine treatment, suggesting less activated endothelial cells and limited infiltration of leukocytes to the aortic wall and in support of the notion that nicotine targets the CAP, at least initially. However, other mechanisms affected by nicotine may result in the augmented AAAs seen in the present study. In non-aneurysmal abdominal aorta, the main cell type in the aortic wall is VSMC. It is well described that a phenotypic switch from contractile to a synthetic and proliferative form of VSMCs contributes to AAA expansion [22]. Upregulation of the Ras signaling pathway, which has been associated with the switch from contractile VSMC to the synthetic and proliferative phenotype [42], could perhaps overrule the anti-inflammatory effect of nicotine mediated by CAP. It has already been shown in vitro that nicotine exposure directly drives human VSMCs from a contractile to synthetic-like phenotype [43]. 

Moreover, our proteome data show that MARCKS is significantly downregulated in abdominal aorta after nicotine treatment. MARCKS is a filamentous actin (F-actin)-binding protein that regulates actin assembly dynamics during cell migration supporting lamellipodia and filopodia formation. MARCKS expression has been shown to be involved in vascular remodeling and promoting VSMC migration, resulting in neointimal hyperplasia after artery injury, which can be prevented by knocking down MARCKS [44,45]. Interestingly, knock down of MARCKS in endothelial cells augmented reendothelialization after vascular injury [45]. During induction of AAA formation in our rats, the endothelial layer become severely damaged; thus, a downregulation of MARCKS reduces the formation of an intact endothelial layer that could eventually lead to increased AAA formation. Downregulation of MARCKS in aortic VSMCs dampens the phenotypic shift to a more synthetic and migratory phenotype, which is one of the initial steps in AAA expansion, supporting an initial protective effect of AAA formation. 

We also found downregulation of the cytoskeletal and plasma membrane-bound Palm in our proteomic analysis of non-aneurysmal abdominal aorta. This protein is involved in lymphangiogenesis, thereby supporting cancer growth [46]. Thus, these data support a protective initial effect in AAA development and is in line with the findings that nicotine has an anti-inflammatory response and an anti-ROS effect in non-aneurysmal aorta. The absence of Palm dampens lymphatic endothelial cell adhesion and migration, thereby limiting tumor growth [46]. In cardiovascular diseases, increased lymphangiogenesis provides a drainage of cholesterol reverse transport, cytokines, and macrophages [47]; thus, a less-developed lymph network may contribute to augmented AAA expansion. This shift in VSMC phenotype was not observed at our aneurysmal mRNA levels of either contractile α-actin, myocardin (a guardian of the contractile and non-inflammatory VSMC phenotype [48]), nor of the marker for the synthetic phenotype osteopontin, where no change between groups was observed.

Furthermore, the existing literature reports conflicting findings on the immunoregulatory function of α7nAChR signaling [22,49,50]; thus, its role in cardiovascular diseases remains enigmatic [51]. Ulleryd et al. demonstrated that activation of α7nAChRs with a selective α7nAChRs agonist decreases atherosclerosis in ApoE^−/−^ mice with suppression of pro-inflammatory cytokines as a possible athero-protective mechanism [51]. Furthermore, Hashimoto et al. found that a selective α7nAChRs agonist attenuates atherogenesis and AAA formation in ApoE^−/−^ mice, supporting the anti-inflammatory capabilities of α7nAChRs [52]. Meanwhile, Wang et al. [53] tested a higher dose of nicotine that led to accelerated atherosclerosis in ApoE^−/−^ mice by activating α7nAChRs in mast cells, supporting the pro-inflammatory effects of α7nAChRs. Thus, the data indicate that selective activation of α7nAChRs in different immune types result in pro-inflammatory properties in mast cells and dendritic cells, while activation of α7nAChRs in macrophages is associated with anti-inflammatory properties.

In our study, we found no significant infiltration of macrophages or neutrophils, nor did we find that the aneurysmal mRNA levels of TNF, IL6 or IL10 were affected by nicotine treatment. One reason for the lack of effect on inflammatory markers could be the time of analyses, as the pro-inflammatory response in this model was more pronounced in the first 2 weeks of AAA development [54,55]. However, because our primary outcome was differences in AAA size which needs 21-28 days to reach maximal aortic diameter, this is the reason for choosing this time point. Lui et al. [56] did not detect any effect in inflammatory cytokine mRNA levels, aortic wall thickness or blood pressure after long-term exposure to nicotine in drinking water in lean rats, whereas nicotine exposure in obese rats on a high fat diet increased systolic blood pressure, inflammation, aortic superoxide production and endothelial dysfunction with impaired responsiveness to acetylcholine-mediated vasodilatation. As our AAA model was induced in lean rats on normal chow, this could explain the lack of an impact on inflammation. 

Another explanation that nicotine promotes AAA growth rather than prevents it could be the chosen dose of nicotine used in this study. It is well recognized that the effects of nicotine act in a dose-dependent manner, with toxic cardiovascular effects at higher doses [57]. This leaves us with the question of whether the chosen nicotine dose of 1.25 mg/kg/day was too high. If this dose were to be converted into a human dose, it would correspond to 0.203 mg/kg, resulting in a total of 16.2 mg nicotine for a human weighing 80 kg [58]. Inhalation of one cigarette results in a 0.5–2 mg uptake of nicotine [59]. Thus, the chosen dose corresponds to smoking around eight cigarettes a day, which might be above the intended low-dose administration of nicotine. Our dose is, however, in line with the one used by Han et al., who demonstrated that nicotine administration of 1.0 mg/kg/day exerted anti-inflammatory effects in a murine model [15]. However, the half-life of plasma nicotine in mice is shorter than in rats, necessitating the use of a higher dose of nicotine in mice compared to rats [32]. Furthermore, pretreatment with nicotine (1.0 mg/kg) protected against inflammatory and reactive oxygen species damage in both the kidney and liver after ischemic reperfusion damages in rats [60,61], suggesting an appropriate level for our chosen dose. 

It is well recognized that nAChRs are characterized by their rapid desensitization, leading to the development of acute tolerance [62]. When nicotine is continuously present, nAChRs can transition to an inactive state where they cannot be reactivated by nicotine [62], thus halting the beneficial effects of nicotine through its receptor. If the dose of an agonist for nAChRs becomes too high, desensitization can occur [63]. A lower nicotine dose (< 1.0 mg/kg/day) could therefore have been more adequate in our study to achieve anti-inflammatory effects. Furthermore, because nicotine is not a selective agonist for α7nAChRs, it also activates other homodimeric nAChRs or heteromeric nAChRs, as well as homomeric nAChRs, such as α-4b2-, α-6b2- and α-3b4-nicotine acetylcholine receptor [64], which are also present in the rat aorta [65]. As such, we cannot ignore that our nicotine treatment may act via other receptors than α7nAChRs. That being said, in vitro findings of the vasodilatory properties of nicotine are mediated via α7nAChR signaling in both normotensive and hypertensive rats [65]. However, Watanabe et al. suggests that increase in AAA size by nicotine is mediated by receptors other than α7nAChRs [38]. In fact, Ren et al. demonstrated that nicotine promotes atherosclerosis development mediated by α1nAChRs-induced activation of the calpain-1/MMP-2/MMP-9 signaling pathway [66]. 

In our study, nicotine treatment reduced aneurysmal MMP9 activity, although there was no obvious effect on elastin integrity, nor were there any differences in MMP9-positive cells in the aneurysm wall, even though nicotine treatment led to the development of larger AAAs. It is generally accepted that elevated MMP activity contributes to elastin degradation and destruction of the aortic wall integrity, consequently facilitating the progression of AAAs [57]. The reason for this discrepancy is still unclear. In healthy mice, long-term exposure of a high nicotine dose (25 mg/kg/day) resulted in thinning and fragmentation of elastin in the abdominal aortic wall [67]. One explanation for our results could be that aneurysm expansion reached its maximum around day 21; thus, we cannot rule out that augmented MMP activity is not present at an earlier stage of AAA expansion. In support of our findings, low-dose nicotine treatment (0.4 mg/kg) twice daily for a week also resulted in decreased skeletal muscular MMP activity and the inflammatory cytokines decreased in mdx male mice that are prone to develop inflammatory myopathy [68]. Furthermore, others have shown in a mouse model of osteoarthritis and in LPS-stimulated bone-marrow-derived macrophages that nicotine treatment in a concentration like ours (1.0 mg/kg) can reduce MMP9 [69]. Another possibility for augmented AAA progression in the nicotine-treated group could be caused by a nicotine-mediated elevation in systolic blood pressure, as seen after long-term exposure in obese rats [56]. We did not measure blood pressure in our study, but based on heart to body weight ratio, there were no differences in size between groups, suggesting no differences in blood-pressure-mediated ventricular hypertrophy. Thus, major differences in blood pressure are less likely. 

This study has some limitations. Despite several pathological similarities to human AAAs, the elastase AAA model in rats does not imitate the exact pathologic conditions in human AAAs [70]. The acute nature of the aortic injury in this AAA model leads to rapid onset and limited duration of aortic expansion, which is different from human AAAs [7]. Thus, difficulties in translating the effects of nicotine on rat AAAs to clinical use in human AAAs is another limitation of this study. 

## 5. Conclusions

In conclusion, nicotine at a dose of 1.25 mg/kg/day augments AAA expansion in this elastase AAA model by an undetermined mechanism, which is not related to changes in elastin integrity, nor the infiltration of neutrophils or macrophages. However, proteomic data support anti-inflammatory effects of nicotine in non-aneurysmal abdominal aorta. These results do not support the notion that low-dose nicotine administration could be used for the prevention of AAA progression.

## Figures and Tables

**Figure 1 biomedicines-11-01417-f001:**
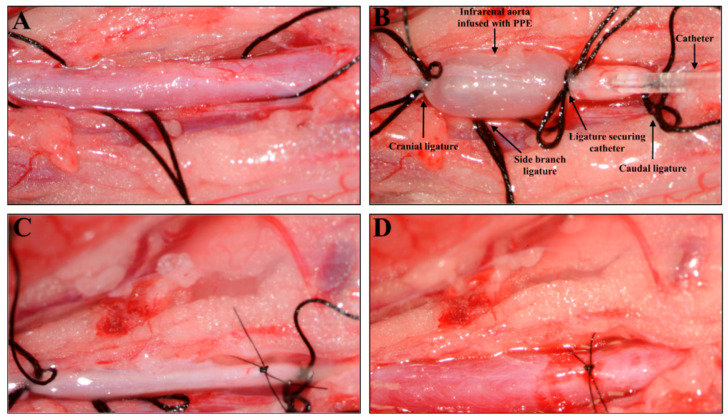
Surgical induction of an abdominal aortic aneurysm with intraluminal infusion of porcine pancreatic elastase (PPE) in a Sprague–Dawley rat. (**A**) Isolation of the infrarenal aorta before ligation and infusion with PPE. (**B**) Infrarenal aorta under ligation and intraluminal PPE infusion. (**C**) Infrarenal aorta after infusion and closure of the incision with an 8.0 suture. (**D**) Infrarenal aorta with restored blood flow after removal of ligatures.

**Figure 2 biomedicines-11-01417-f002:**
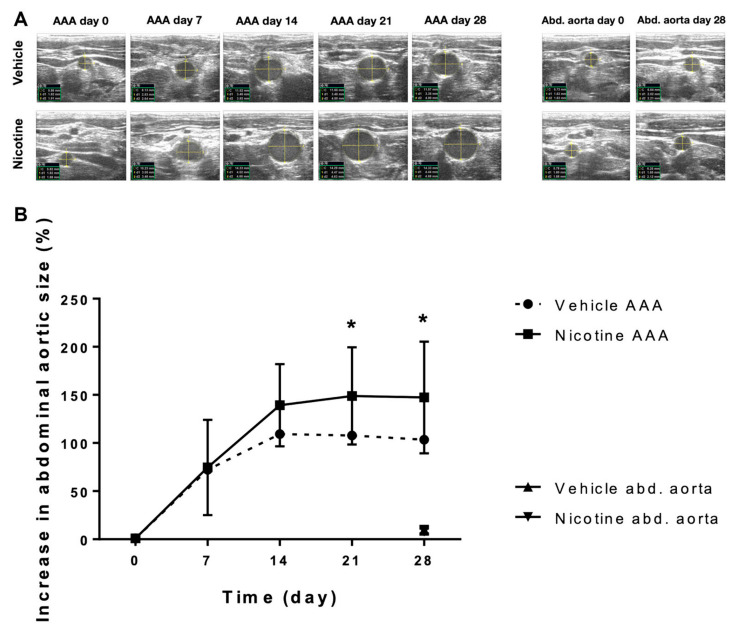
The effect of nicotine treatment on abdominal aortic aneurysm progression. (**A**) Representative ultrasound measurements of the baseline aortic size at expected aneurysm site prior to surgical aneurysm induction and post-surgery aortic size measurements at day 7, 14, 21 and 28 in the vehicle and nicotine groups. The most proximal part of the infrarenal aorta, just distal from the exit point of the left renal artery, was used as reference for the increase in aortic size during the experimental period of 28 days (Abd. aorta). (**B**) The relative increase in abdominal aortic aneurysm size measured by ultrasound recordings from day 0–28 in the vehicle group (*n* = 14) and the nicotine group (*n* = 13). Values are presented as mean ± standard deviation. * *p* < 0.05. Abbreviations in ultrasound images: C = circumference; d1 = anterior–posterior diameter; d2 = horizontal diameter; AAA = abdominal aortic aneurysm; Abd. aorta = abdominal aorta.

**Figure 3 biomedicines-11-01417-f003:**
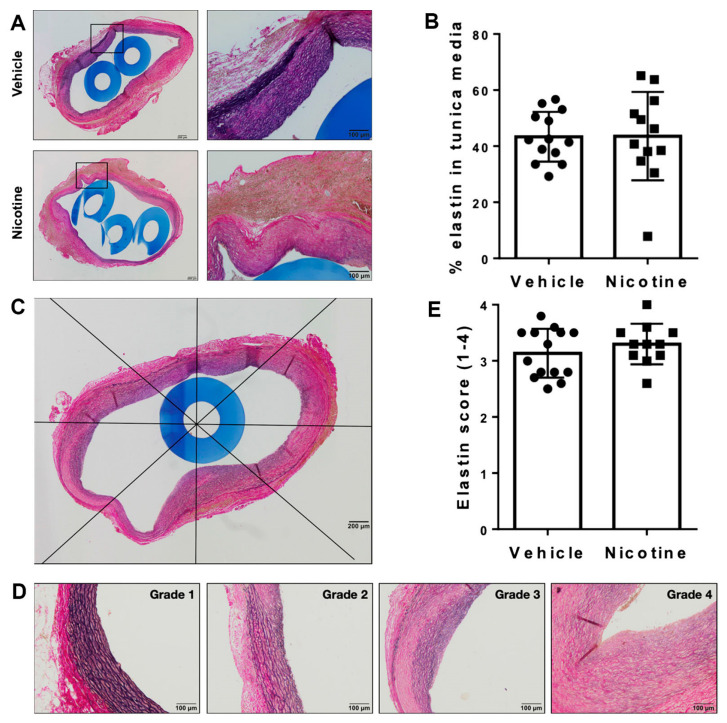
Assessment of elastin in aneurysmal tissue. (**A**) Representative histological images of aneurysmal cross-sections stained with Miller’s elastin to assess elastin in tunica media on day 28 in vehicle and nicotine group. Boxes on the left micrographs are shown at a higher magnification in the right micrographs. (**B**) Percentage of elastin remaining in tunica media on day 28 in vehicle and nicotine group (*n* = 13/12). (**C**) Representative image of a cross-sectional abdominal aortic aneurysm in a rat treated with nicotine stained with Miller’s elastin and divided into 8 areas. The score of elastin degradation was assessed in each area. The mean value of each cross-section was used for statistical analysis. (**D**) Representative images of the grades of elastin degradation (1 = preserved elastin architecture; 4 = total disruption of complete concentric elastin lamellae). (**E**) Average score of elastin degradation in tunica media in vehicle and nicotine group (*n* = 14/11).

**Figure 4 biomedicines-11-01417-f004:**
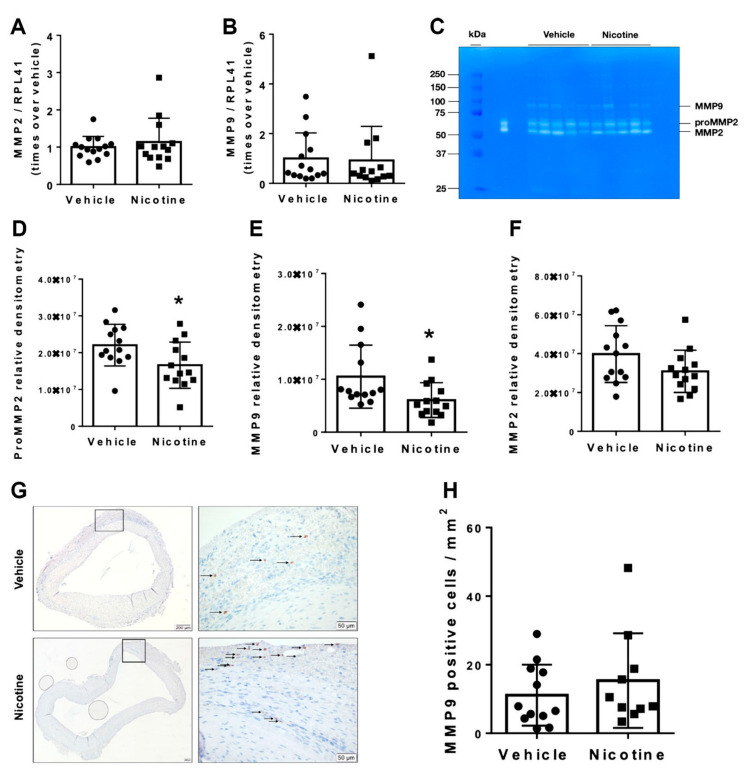
Messenger RNA (mRNA) levels and activity of matrix metalloproteinases (MMPs) in aneurysmal tissue normalized to Ribosomal Protein L41 (RPL41). mRNA levels of (**A**) MMP2 (*n* = 14/13) and (**B**) MMP9 (*n* = 14/13) in aneurysmal tissue on day 28 in vehicle and nicotine group. (**C**) Representative image of gelatin zymography to detect MMP activity in abdominal aortic aneurysm in vehicle and nicotine group. The bands showed pro-MMP2 (72 kDa), MMP2 (67 kDa) and MMP9 (78 kDa) activity. The densities of the zymographic bands were quantified by densitometry for (**D**) MMP2 (*n* = 13/13), (**E**) pro-MMP2 (*n* = 13/13) and (**F**) MMP9 (*n* = 13/13). (**G**) Representative micrographs of MMP9 staining from aneurysmal tissue in vehicle and nicotine groups. Arrows indicate positive cell stains. (**H**) Number of MMP9-positive cells per mm^2^ in the two groups. Values are presented as mean ± standard deviation. * *p* < 0.05.

**Figure 5 biomedicines-11-01417-f005:**
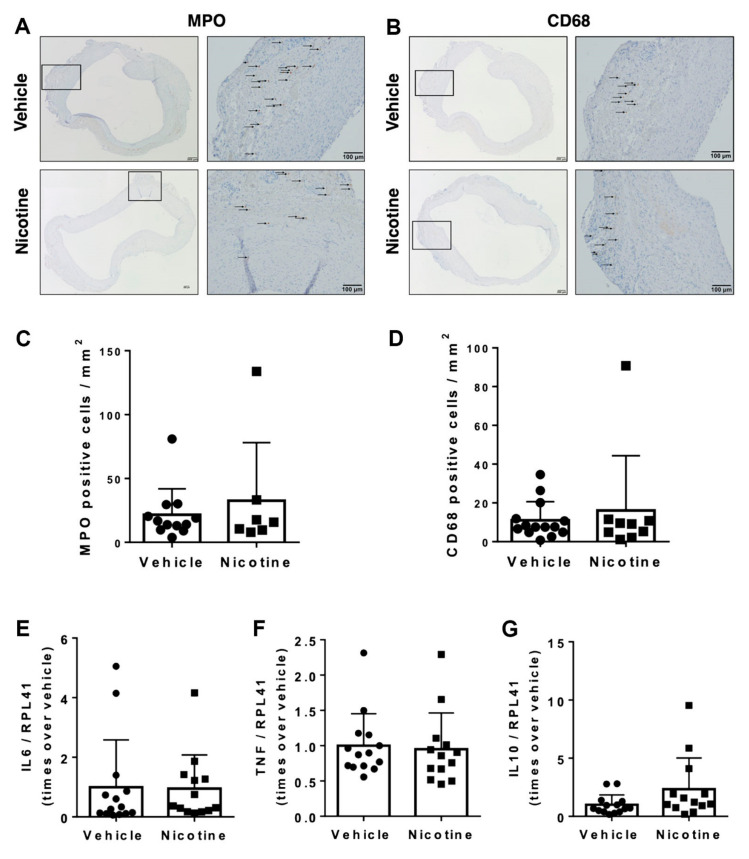
Infiltration of inflammatory cells in abdominal aortic aneurysms. Representative micrographs of (**A**) myeloperoxidase (MPO) as a marker for neutrophils and (**B**) Cluster of Differentiation 68 (CD68) as a marker for macrophages from aneurysmal tissue in vehicle and nicotine group. Arrows indicate positive cell stains. Quantity of (**C**) MPO (*n* = 12/7) and (**D**) CD68 (*n* = 14/9) positive cells per mm^2^ in each group. Messenger RNA (mRNA) levels of (**E**) IL6 (*n* = 14/13), (**F**) TNF (*n* = 14/13) and (**G**) IL10 (*n* = 14/13) in aneurysmal tissue on day 28 in the groups. All mRNA levels were normalized to Ribosomal Protein L41 (RPL41).

**Figure 6 biomedicines-11-01417-f006:**
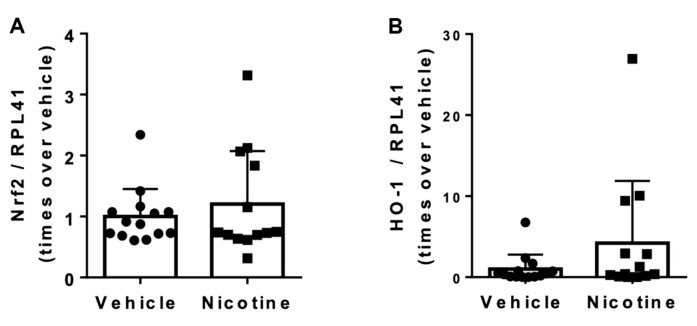
The effect of nicotine on aneurysmal oxidative stress at post-surgery day 28. (**A**) Messenger RNA (mRNA) levels of nuclear factor erythroid 2-related factor 2 (Nrf2) (*n* = 14/13) and (**B**) mRNA levels of heme oxygenase (HO)-1 (*n* = 14/13) in the aneurysmal tissue normalized to Ribosomal Protein L41 (RPL41). Values are presented as median ± interquartile range.

**Figure 7 biomedicines-11-01417-f007:**
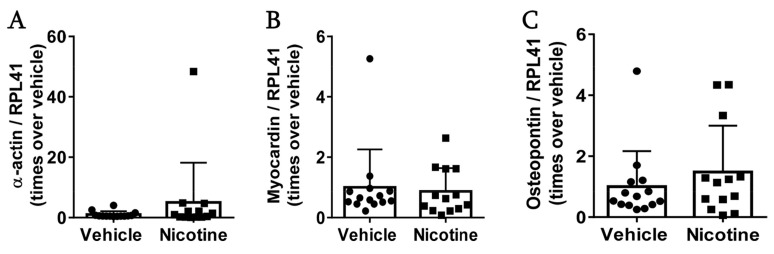
The effect of nicotine on smooth muscle cells at post-surgery day 28. Messenger RNA (mRNA) levels of (**A**) α-actin (*n* = 14/13), (**B**) myocardin (*n* = 14/13) and (**C**) osteopontin (*n* = 14/13) in the aneurysmal tissue. All data were normalized to Ribosomal Protein L41 (RPL41) mRNA levels. Values are presented as median ± interquartile range.

**Figure 8 biomedicines-11-01417-f008:**
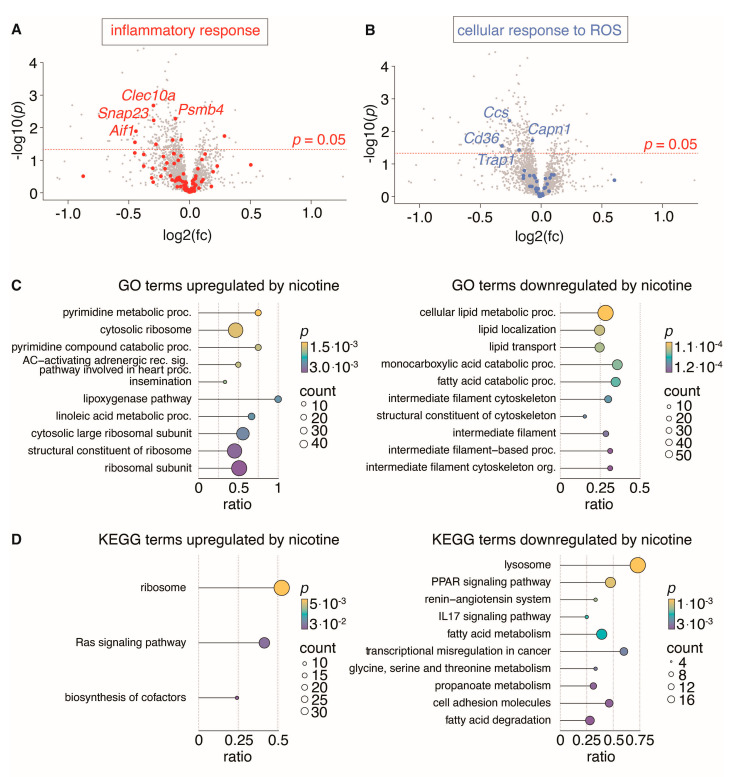
Proteomic analysis of non-aneurysmal aorta segments from rats treated with nicotine (*n* = 12) and untreated rats (*n* = 13). (**A**,**B**): Volcano plots (nicotine-treated rats relative to untreated rats) showing regulation of all identified proteins, with proteins belonging to gene ontology (GO) terms “inflammatory response” (GO:0006954) and “cellular response to ROS” (GO: 0034614) highlighted in red (**A**) and blue (**B**) with the four most significantly regulated proteins labeled. (**C**,**D**): Dot plots show the top 10 enriched biological process GO terms (**C**) and KEGG terms (**D**) either up- or downregulated by nicotine. Dot size indicates number of proteins found to be regulated by nicotine for a given term and dot color indicates the level of nominally significance. The protein ratio shows the proportion of a given term covered by proteins regulated by nicotine.

**Table 1 biomedicines-11-01417-t001:** Organ-to-body weight ratios between vehicle group and nicotine group.

Organ Weight/Body Weight (mg/g)	Vehicle (*n* = 14)	Nicotine (*n* = 13)	*p*-Value
Heart	3.33 ± 0.38	3.27 ± 0.41	0.437
Spleen	2.75 ± 0.37	2.61 ± 0.64	0.375
Lungs	4.07 ±0.32	3.89 ± 0.16	0.099
Kidneys	6.16 ± 0.70	5.95 ± 0.67	0.437

## Data Availability

The mass spectrometry proteomics data have been deposited to the ProteomeXchange Consortium via the PRIDE [71] partner repository with the dataset identifier PXD040414.

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
