# Peer review of "Nicotine Administration Augments Abdominal Aortic Aneurysm Progression in Rats"

_biomedicines, 2023, doi:10.3390/biomedicines11051417_

Round 1
Reviewer 1 Report (New Reviewer)
I found the article interesting and very interestingly presented. The topic related to abdominal aortic aneurysms is important and up-to-date because we know too little to treat these pathologies effectively.
I find the added proteomic analyses interesting - they are an important supplement and provide interesting results, although they also introduce a bit of confusion. It was very difficult for me to grasp what they actually bring in the context of the hypothesis put forward by the authors.
They generate interesting results, especially for the aorta without aneurysms. It is unclear to me what we learn in the context of nicotine and aneurysms. Proteomics studies were performed on the walls of the abdominal aorta in the absence of aneurysms.
The results section is consistent, and I have no objections here, although the part concerning the results of proteomics research is the least clear. It ends with the statement: no pathways were regulated after correcting for multiple testing. So how should it be properly understood? What about the 3 proteins that post a correction for multiple testing were significantly downregulated?
The authors do not refer to this further in the text of the article.
I have the most objections to the discussion and especially to the conclusions.
I propose to divide the discussion into sections.
In turn, the conclusions are too general; generalizations of the type "we can conclude that nicotine at the given dose augments AAA expansion" should be avoided. This should also be corrected in the abstract where conclusions appear. Because actually, despite many studies conducted with due diligence, the authors, based on imaging studies, show the existence of the effect of nicotine on the enlargement of the aneurysm but do not find the mechanism - perhaps the first paragraph of the discussion should be entered here - it brings much more than the current conclusions.
Also, I don't see a clear target as to whether it is: 'Therefore, we hypothesized that low-dose nicotine, an α7nAChR agonist, impairs AAA progression in rats by activating CAP.'
Is that the goal? - verification of the hypothesis - if so, the written authors' conclusions do not verify the hypothesis. This should be sorted out. In this situation, the title is also not entirely correct.
From minor remarks:
- In the Ethical statements section - the authors wrote, "we only used males in the study" - I suggest replacing them with male rats.
Author Response
I found the article interesting and very interestingly presented. The topic related to abdominal aortic aneurysms is important and up-to-date because we know too little to treat these pathologies effectively.
I find the added proteomic analyses interesting - they are an important supplement and provide interesting results, although they also introduce a bit of confusion. It was very difficult for me to grasp what they actually bring in the context of the hypothesis put forward by the authors.
They generate interesting results, especially for the aorta without aneurysms. It is unclear to me what we learn in the context of nicotine and aneurysms. Proteomics studies were performed on the walls of the abdominal aorta in the absence of aneurysms.
RESPONSE: Thank you for these comments. We reason that since nicotine treatment augments aneurysm development, then proteomic profiling of the aneurysms per se would mainly reflect differences in disease state (large vs small aneurysms) rather than shedding light on the underlying mechanisms driving those differences. To better understand the effect of nicotine, we believe it is crucial to identify the specific changes induced by nicotine that may predispose the vessel wall to aneurysm development. We hope this explanation provides greater clarity on our reason.
The results section is consistent, and I have no objections here, although the part concerning the results of proteomics research is the least clear. It ends with the statement: no pathways were regulated after correcting for multiple testing. So how should it be properly understood? What about the 3 proteins that post a correction for multiple testing were significantly downregulated?
The authors do not refer to this further in the text of the article.
Response: We understand the confusion, and hope that the changes made in the text will help clarify. Indeed, we did find significant regulation of individual proteins after correcting for multiple testing (proteins with an FDR < 0.05; Tropomodulin-2 (Tmod2), Myristoylated alanine-rich C-kinase substrate (Marcks), and Paralemmin-1 (Palm). Then we tested specific hypotheses of the effect of nicotine on the “anti-inflammatory response” and “cellular response to oxygen species” (using Go-Term ontology and χ2 testing). These tests do not require any adjustment since we do not perform “multiple testing” (Figure 8a and B). However, in our exploratory approach, correcting for multiple testing is required, and in doing so, we find no pathways that were significantly regulated (no pathways with q < 0.05 in supplementary tables S2 and S3). However, we still chose to plot the top-ranking pathways, that displayed nominal significance (fig 8C-D) and could potentially be of interest as it is the most regulated pathways.
It has been changed to the following: “To assess the impact of nicotine on the abdominal aortic wall in the absence of aneurysms and potentially reveal mechanisms driving toward larger aneurysm development by nicotine treatment, we conducted proteomic analysis on the abdominal aorta of rats.” page 12 line 362-364
And: ”To test the hypothesis that nicotine treatment had an anti-inflammatory or anti-reactive oxidative species effect on the aortic wall, we investigated gene ontology terms “inflammatory response” (GO:0006954) and “cellular response to reactive oxygen species” (GO:0034614) (Fig. 8A-B).” page 12 line 374-377
And: ”When corrected for multiple testing, no pathways were significantly regulated (q<0.05, Supplementary Tables S2 and S3). However, several pathways were nominally regulated in nicotine-treated rat aortas for example the lipoxygenase pathway and pyrimidine metabolic processes, and Ras signaling pathway is amongst the upregulated pathways, while fatty acid metabolism, cell adhesion, IL17 signaling pathway, and the renin-angiotensin system is downregulated that could potentially contribute to the understanding of the nicotine effect on abdominal aorta (Fig. 8C-D).” page 13 line 389-395.
As for the 3 proteins that were significantly downregulated in nicotine-treated non-aneurysmal abdominal aorta we have not included them in the discussion:
“Moreover, our proteome data shows that MARCKS is significantly downregulated in the abdominal aorta after nicotine treatment. MARCKS is a filamentous actin (F-actin) binding protein that regulates actin assembly dynamics during cell migration supporting lamellipodia and filopodia formation. MARCKS expression has been shown to be involved in vascular remodeling and promoting VSMC migration resulting in neointimal hyperplasia after artery injury, which can be prevented by knocking down MARCKS (44, 45). Interestingly, knockdown of MARCKS in endothelial cells augmented reendothelialization after vascular injury (45). During induction of AAA formation in our rats, the endothelial layer gets severely damaged, thus downregulation of MARCKS will reduce the formation of an intact endothelial layer that could eventually lead to increased AAA formation. Downregulation of MARCKS in aortic VSMC will dampen the phenotypic shift to a more synthetic and migratory phenotype, which is one of the initial steps in AAA expansion, supporting an initial protective effect of AAA formation. We also found downregulation of the cytoskeletal and plasma membrane-bound Palm in our non-aneurysmal abdominal aorta proteomic analysis. This protein is involved in lymphangiogenesis, thereby supporting cancer growth (46). Thus, these data support a protective initial effect in AAA development and are in line with the findings that nicotine has an anti-inflammatory response and anti-ROS effect in the non-aneurysmal aorta. The absence of Palm dampens lymphatic endothelial cell adhesion and migration, thereby limiting tumor growth (46). In cardiovascular diseases increased lymphangiogenesis, provide drainage of cholesterol reverse transport, cytokines, and macrophages (47), thus a less developed lymph network may contribute to augmented AAA expansion.” page 13 line 429-450.
I have the most objections to the discussion and especially to the conclusions.
I propose to divide the discussion into sections.
In turn, the conclusions are too general; generalizations of the type "we can conclude that nicotine at the given dose augments AAA expansion" should be avoided. This should also be corrected in the abstract where conclusions appear. Because, despite many studies conducted with due diligence, the authors, based on imaging studies, show the existence of the effect of nicotine on the enlargement of the aneurysm but do not find the mechanism - perhaps the first paragraph of the discussion should be entered here - it brings much more than the current conclusions.
RESPONSE: We agree that the conclusion was too general, and we have now rephrased the conclusion throughout the manuscript: “In conclusion, nicotine at a dose of 1.25 mg/kg/day augments AAA expansion in this elastase AAA model, by a mechanism undetermined, but not related to changes in elastin integrity, nor infiltration of neutrophils nor macrophages. Although proteomic data support the anti-inflammatory effects of nicotine in the non-aneurysmal abdominal aorta. These results do not support that low dose nicotine administration could be used for prevention of AAA progression.“ page 16 lines 542-547.
Also, I don't see a clear target as to whether it is: 'Therefore, we hypothesized that low-dose nicotine, an α7nAChR agonist, impairs AAA progression in rats by activating CAP.'
Is that the goal? - verification of the hypothesis - if so, the written authors' conclusions do not verify the hypothesis. This should be sorted out. In this situation, the title is also not entirely correct.
RESPONSE: We agree it is a bit misleading that we hypothesize that low-dose nicotine works directly through α7nAChR and activates CAP, as we do not test this partway directly. We have modified our hypothesis to the following: “In this study, we hypothesize that low-dose nicotine impairs the progression of elastase-induced AAAs in male rats by exerting anti-inflammatory and anti-oxidative stress properties.”
Changes were made in the abstract on page 1 lines 18-19 and at the end of the introduction page 2 line 84-86.
From minor remarks:
- In the Ethical statements section - the authors wrote, "We only used males in the study" - I suggest replacing them with male rats.
RESPONSE: Thank you for your comment. It has now been added to the ethical statement: ”…we only used male rats in this study” page 3 line 92.
Reviewer 2 Report (New Reviewer)
The study presented by Hazikadunic H et al has investigated the role of nicotine administration on abdominal aortic aneurysm expansion and degeneration of matrix preoteinases and inflammatory cytokine expression in the aortic wall in a aortic aneurysm rat model. The data incicate that nicotine administration seems to induce aneurysm progression. However, nicotine reduced MMP2 and 9 activity, although elastin content and elastin degradation was not affected.
The manuscript is of interest and indicates, that the results reported so far are highly contradictory and do not support the idea that nicotine could be used for prevention of aneurysm progression as reported by others.
However, there are several points that are difficult to compare between the different studies: different aneurysm models, different dosages, the point regarding aneurysm induction or progression, the induction of other pathways by nicotine. Although some of these points are already mentioned in the discussion, some should be clarified.
1. It seems that the findings reported support the known risk of aneurysm formation or progression in smokers compared to nonsmokers.
2. Regarding the dose of nicotine used: did the authors have performed other experiments with lower or higher doses of nicotine to see dose-related effects for example on inflammtory responses or short term changes on aortic wall matrix proteinases or elastin degradation ?
3. Some studies report effects caused by nicotine on immune and inflammatory cells on arteriosclerosis. Did the authors have used specific nicotine antgonist or more specific antagonists for a7nAChRs to show that the effects are mediated by this pathway.
4. The point of receptor down regulation is another major point for interpretation of the results. Therefore, are data with this approach available performed in rats having mutant a7nAChRs negative cells ?
5. In the conclusion (section 5., page 15) the authors should mention their elastase AAA rat model in the first sentence. In addition, the results presented do not support the recommendation for low dose nicotine administration as possible treatment for prevention of AAA progression.
6. Furthermore, the last part in the conclusion (5., page 15) mentions other opinions (literature), suggesting that targeting the a7nAChR signalling could be a potential treatment option against AAA expansion and rupture. However, this idea is not supported by the data presented here and should not be part of the conclusion.
Author Response
The study presented by Hazikadunic H et al has investigated the role of nicotine administration on abdominal aortic aneurysm expansion and degeneration of matrix preoteinases and inflammatory cytokine expression in the aortic wall in a aortic aneurysm rat model. The data incicate that nicotine administration seems to induce aneurysm progression. However, nicotine reduced MMP2 and 9 activity, although elastin content and elastin degradation was not affected.
The manuscript is of interest and indicates, that the results reported so far are highly contradictory and do not support the idea that nicotine could be used for prevention of aneurysm progression as reported by others.
However, there are several points that are difficult to compare between the different studies: different aneurysm models, different dosages, the point regarding aneurysm induction or progression, the induction of other pathways by nicotine. Although some of these points are already mentioned in the discussion, some should be clarified.
RESPONSE: We have tried in our discussion to discuss the complexity of comparing different models and doses etc. We hope you find the updated version clearer.
- It seems that the findings reported support the known risk of aneurysm formation or progression in smokers compared to nonsmokers.
RESPONSE: Yes, our data supports the known risk of smoking (nicotine administration) on AAA progression compared to non-smokers. The aim of this study was however, to test if a low-dose nicotine administration would result in anti-inflammatory and anti-oxidative stress described by others and the recent data from the DANCAVAS study, that could protect against AAA progression. The effect seen here by our proteome analyses supports an anti-inflammatory and anti-oxidative stress effect in the non-aneurysmal abdominal aorta. While nicotine administration at the chosen dose to the damaged abdominal aorta (1 day after elastase infusion to initiate AAA development) facilitate an augmented AAA progression by a mechanism undetermined, but not related to changes in elastin integrity, nor infiltration of neutrophils nor macrophages.
- Regarding the dose of nicotine used: did the authors have performed other experiments with lower or higher doses of nicotine to see dose-related effects for example on inflammtory responses or short term changes on aortic wall matrix proteinases or elastin degradation?
RESPONSE: It is a great suggestion, but unfortunately, we did not test different doses either as an additional group or as short-term experiments. It could have been of relevance to test nicotine in vitro in either bone marrow-derived macrophages or vascular smooth muscle cells to test several concentrations of nicotine and their role in MMP production and their inflammatory response by measuring cytokine production. However, due to the limited time allowed for the resubmission (10 days) we were unable to conduct additional experiments to clarify this.
- Some studies report effects caused by nicotine on immune and inflammatory cells on arteriosclerosis. Did the authors have used specific nicotine antagonist or more specific antagonists for a7nAChRs to show that the effects are mediated by this pathway.
RESPONSE: Yes, we agree this would have been fruitful to examine. We did unfortunately not include an additional arm in our aneurysm study and thereby testing the direct effect of activating or blocking the α7nAChRs. Also, it would have been of interest to test it in in vitro settings described above. However, the time limitation for this resubmission made it impossible to perform additional experiments. It appears that nicotine acting via α7nAChRs is cell type-specific. To further specify the cell-specific activation of α7nAChRs, we have included in the discussion that “…data indicate that selective activation of α7nAChRs in different immune types result in pro-inflammatory properties in mast cells and dendritic cells while activation of α7nAChRs in macrophages associates with anti-inflammatory properties.” page 14 line 465-467.
- The point of receptor down regulation is another major point for interpretation of the results. Therefore, are data with this approach available performed in rats having mutant a7nAChRs negative cells?
RESPONSE: As far as we are aware of there are no available rat strain that has a mutant in α7nAChRs, although it is an important point, it will be hard to test. There are data from mice that lack α7nAChRs in their hematopoietic cells. These mice provide inconsistent effects on atherogenesis, were both no effect Kooijman et al. (doi: 10.1111/jth.12765) while Johansson et al. (doi: 10.1161/ATVBAHA.114.303892) detected an augmented atherogenesis. This again points to the complex condition involving many cell types where activation of α7nAChRs might have opposing effects. As included in the discussion and provided above.
- In the conclusion (section 5., page 15) the authors should mention their elastase AAA rat model in the first sentence. In addition, the results presented do not support the recommendation for low dose nicotine administration as possible treatment for prevention of AAA progression.
RESPONSE: Thank you for this comment. It has now been added to the conclusion: “In conclusion, nicotine at a dose of 1.25 mg/kg/day augments AAA expansion in this elastase AAA model…” page 16 line 542-543.
- Furthermore, the last part in the conclusion (5., page 15) mentions other opinions (literature), suggesting that targeting the a7nAChR signalling could be a potential treatment option against AAA expansion and rupture. However, this idea is not supported by the data presented here and should not be part of the conclusion.
RESPONSE: Yes, we agree that our data does not support the last sentence and it has therefore been removed.
Round 2
Reviewer 1 Report (New Reviewer)
All my comments were taken into account correctly. I have no more comments. The article has been appropriately corrected and in my opinion, is ready to be accepted and published
Reviewer 2 Report (New Reviewer)
the authors have now improved their manuscript based on the recommendations given by the reviewers.
This manuscript is a resubmission of an earlier submission. The following is a list of the peer review reports and author responses from that submission.
Round 1
Reviewer 1 Report
The authors demonstrated nicotine administration promotes AAA expansion, although its mechanism are still unclear. They discuss the association between AAA and alpha-7cAChR or CAP; however, they do not assess it in this paper. I have comments to improve this paper.
1. The authors should assess female AAA to show the trends are similar to or different from male AAA, even if it is preliminary data.
2. How did the authors confirm that nicotine was successfully administrated. Should they measure blood nicotine concentration?
3. The author only assessed organ weights as an adverse effect of nicotine administration. Blood analysis such as BUN, Creatinine, AST, ALT, should be needed to exclude organ failure.
4. Nicotine can affect blood pressure and heart rate. Please show data of blood pressure and heart rate during nicotine administration.
5. Why nicotine promotes AAA expansion, regardless of MMP-9 was decreased?
6. The authors should assess the associations between MMP decrease and CAP.
7. How did nicotine administration affect CAP activation?
8. The authors should assess inflammation within 14days after AAA creation.
9. The authors show many negative data. It is ok, but why do not they perform RNA or protein array to seek factors which may be associated with AAA progression and nicotine?
Reviewer 2 Report
Dear Authors,
although the study is presented very nicely and scientifically sound, the study question and results - adverse effect of (relatively high dose) nicotine on AAA development - lack novelty and do not provide significant promotion of current knowledge.